# Developing an Advanced PM$_{2.5}$ Exposure Model in Lima, Peru

**Bryan N. Vu [1], Odón Sánchez [2] , Jianzhao Bi [1], Qingyang Xiao [1], Nadia N. Hansel [3], William Checkley [3], Gustavo F. Gonzales [4,5,6], Kyle Steenland [1] and Yang Liu [1],\***

[1] Department of Environmental Health, Rollins School of Public Health, Emory University, Atlanta, GA 30322, USA; bryan.vu@emory.edu (B.N.V.); jianzhao.bi@emory.edu (J.B.); xiaoqy@planet-data.cn (Q.X.); nsteenl@emory.edu (K.S.)

[2] Carrera Profesional de Ingeniería Ambiental, Universidad Nacional Tecnológica de Lima Sur (UNTELS), cruce Av. Central y Av. Bolivar, Villa El Salvador, Lima 15102, Peru; osanchezbr@gmail.com

[3] Division of Pulmonary and Critical Care, School of Medicine, Johns Hopkins University, Baltimore, MD 21205, USA; nhansel1@jhmi.edu (N.N.H.); wcheckl1@jhmi.edu (W.C.)

[4] Endocrinology and Reproduction Unit, Research and Development Laboratories (LID), Faculty of Sciences and Philosophy, Universidad Peruana Cayetano Heredia, Lima 15102, Peru; gustavo.gonzales@upch.pe

[5] Department of Biological and Physiological Sciences, Faculty of Sciences and Philosophy, Universidad Peruana Cayetano Heredia, Lima 15102, Peru

[6] Instituto de Investigaciones de la Altura, Universidad Peruana Cayetano Heredia, Lima 15102, Peru

\* Correspondence: yang.liu@emory.edu

**Abstract:** It is well recognized that exposure to fine particulate matter (PM$_{2.5}$) affects health adversely, yet few studies from South America have documented such associations due to the sparsity of PM$_{2.5}$ measurements. Lima's topography and aging vehicular fleet results in severe air pollution with limited amounts of monitors to effectively quantify PM$_{2.5}$ levels for epidemiologic studies. We developed an advanced machine learning model to estimate daily PM$_{2.5}$ concentrations at a 1 km$^2$ spatial resolution in Lima, Peru from 2010 to 2016. We combined aerosol optical depth (AOD), meteorological fields from the European Centre for Medium-Range Weather Forecasts (ECMWF), parameters from the Weather Research and Forecasting model coupled with Chemistry (WRF-Chem), and land use variables to fit a random forest model against ground measurements from 16 monitoring stations. Overall cross-validation R$^2$ (and root mean square prediction error, RMSE) for the random forest model was 0.70 (5.97 μg/m$^3$). Mean PM$_{2.5}$ for ground measurements was 24.7 μg/m$^3$ while mean estimated PM$_{2.5}$ was 24.9 μg/m$^3$ in the cross-validation dataset. The mean difference between ground and predicted measurements was $-0.09$ μg/m$^3$ (Std.Dev. = 5.97 μg/m$^3$), with 94.5% of observations falling within 2 standard deviations of the difference indicating good agreement between ground measurements and predicted estimates. Surface downwards solar radiation, temperature, relative humidity, and AOD were the most important predictors, while percent urbanization, albedo, and cloud fraction were the least important predictors. Comparison of monthly mean measurements between ground and predicted PM$_{2.5}$ shows good precision and accuracy from our model. Furthermore, mean annual maps of PM$_{2.5}$ show consistent lower concentrations in the coast and higher concentrations in the mountains, resulting from prevailing coastal winds blown from the Pacific Ocean in the west. Our model allows for construction of long-term historical daily PM$_{2.5}$ measurements at 1 km$^2$ spatial resolution to support future epidemiological studies.

**Keywords:** PM$_{2.5}$; air pollution; MAIAC AOD; WRF-chem; random forest; machine learning; remote sensing; Lima; Peru

## 1. Introduction

PM$_{2.5}$ (fine particles with aerodynamic diameter of 2.5 μm or less), is emitted from a large variety of sources including industry, power generation, engine combustion, biomass burning, and natural sources such as sea spray aerosols and wind-blown dust particles [1,2]. PM$_{2.5}$ contributes to 4.2 million global deaths in 2016, and studies have linked exposure to PM$_{2.5}$ with increased adverse health outcomes including respiratory and cardiovascular diseases among not only adults, but also children from North America, Europe, and Asia [3–6]. However, there is a limited number of air pollution studies in South America, where industrialization and continual urban growth may contribute to air pollution levels that far exceed those of Europe and North America [7,8]. Current studies on air pollution in South America pertain mostly to PM$_{10}$ (particles with aerodynamic diameter of 10 μm) or ozone, and are conducted in Brazil, Colombia, and Argentina [8–16]. To date, there has been little to no studies that investigate health outcomes with fine scale exposure measurements in South America.

Lima, Peru is the third-most populous and the second-most polluted major city in the Americas [4]. Lima's air pollution stems from an aging fleet of public transportation in urban areas and the widespread use of indoor biomass stoves in rural areas [4,5]. A report by Banco Bilbao Vizcaya Argentaria (BBVA) Research indicates that the average age of Lima's vehicular fleet exceeds 15 years for private transport vehicles and 22 years for public transport vehicles [6]. Due to the densely populated urbanization of Lima, traffic congestion and exhaust from an aging motor fleet results in particulate matter levels that exceed the World Health Organization's (WHO) standards (25 μg/m$^3$, 24-h mean) [4,17]. A study by Silva et al. found that for six of the 10 ground PM$_{2.5}$ monitors in Lima, 77% of the days between 2014 and 2015 exceeded the WHO's 24-h standards [18]. Moreover, while only 34% of the total population in Peru use solid fuel, 13% of the urban population and over 95% of the rural population rely on biomass fuel for cooking and heating, resulting in high levels of air pollution not only in urban areas but also in the mountainous rural areas [5]. Air pollution affects not only those living in Lima, but also the workers living in the rural communities in the outskirts of the city, who commute 90 to 180 min into the city for work [17]. Yet, there is a limited number of studies on the association between ambient air pollution and health risks in Lima. More studies are needed to assess the effects of PM$_{2.5}$, and potentially to curtail Lima's air pollution effects via new policies to improve air quality standards.

Many of the studies investigating air pollution in Lima have been cross-sectional in design, with childhood asthma as a popular health outcome [19,20]. To date, there have been no studies of air pollution and chronic disease. The limitations in directly utilizing ground-level air monitoring data in epidemiologic studies include the lack of monitoring stations and lack of daily measurements due to maintenance costs [21]. Recently, satellite remote sensing techniques have proven useful in estimating ground PM$_{2.5}$ concentrations [1]. Satellite remote sensing provide aerosol optical depth (AOD), a dimensionless measure of aerosol light extinction within a column of air on Earth's surface [22]. AOD can be used to estimate ground PM$_{2.5}$ concentrations with broad spatial coverage, expanding the ground monitoring networks into the rural areas where ground measurements are lacking [23]. Most commonly used AOD products are derived from the Moderate Resolution Imaging Spectroradiometer (MODIS) and Multiangle Imaging SpectroRadiometer (MISR) aboard the Earth observing System (EOS) satellites named Terra and Aqua launched by the National Aeronautics and Space Administration (NASA) in 1999 and 2002, respectively [24]. These products have also been widely used in recent studies to estimate PM$_{2.5}$ in southern California, China, and Pittsburgh, Pennsylvania [25–27]. A Multiangle Implementation of Atmospheric Correction (MAIAC) algorithm, using time series analysis and image-based processing techniques to make aerosol retrievals and atmospheric corrections over both dark vegetated land and brighter range of surfaces, can be used to retrieve AOD to achieve stronger correlations with PM$_{2.5}$ [28]. MAIAC AOD have been successfully implemented in estimating PM$_{2.5}$ in the United States, Middle East, and China [28–30].

Implementation of remote sensing techniques have proven successful in China and the United States [1,23]. Using non-MAIAC AOD, Liu et al. compared model fit in a two-stage modeling technique

to estimate $PM_{2.5}$ in the Northeast U.S. with and without AOD, with results indicating that the AOD model ($R^2$ = 0.79) has higher predicting power compared to the non-AOD model ($R^2$ = 0.48) [23]. Xiao et al. conducted a study to estimate ground $PM_{2.5}$ concentrations over the Yangtze River Delta of China using MAIAC AOD and ground measurements from 2013 and 2014 with results showing good fit between ground measurements and prediction estimates (cross-validation (CV) $R^2$ = 0.81 for 2013 and 0.73 for 2014) [1]. Additionally, Liang et al. implemented MAIAC AOD to estimate daily $PM_{2.5}$ concentrations in Beijing at 1 $km^2$ spatial resolution with high accuracy (mean annual $R^2$ from 0.79 to 0.86) [31]. The studies listed above found that the correlation between $PM_{2.5}$ and satellite MAIAC AOD, derived from statistical models including generalized linear regression and generalized additive modeling, is greatly improved when land use and meteorological parameters are included; nonetheless, results such as these suggests that MAIAC AOD by itself is a strong predictor of $PM_{2.5}$ concentrations [23,28].

To date, remote sensing techniques have not been utilized in air pollution research in Lima, Peru due to insufficient ground monitoring data to correlate and validate model results. However, in recent years, the Servicio Nacional de Meteorología e Hidrología del Perú (SENAMHI) stations from the Ministry of Environment have begun collecting daily concentrations of $PM_{2.5}$ in Lima, Peru. This presents an opportunity to implement satellite remote sensing techniques in building a model to estimate ground-level $PM_{2.5}$ in a region with critically high levels of air pollution and limited number of epidemiological studies to assess its impact on health risks. In this analysis, we build a $PM_{2.5}$ exposure model to estimate daily $PM_{2.5}$ concentrations at 1 $km^2$ spatial resolution in Lima for years 2010 to 2016. This exposure model is derived from satellite MAIAC AOD, simulation data from chemical transport models (CTMs), meteorological fields from a forecast model, and land use parameters. The resulting daily estimates of $PM_{2.5}$ may be used in epidemiologic studies to assess its impact on both cardiovascular and respiratory health outcomes, and potentially support policies that will mitigate air pollution in Lima, Peru.

## 2. Data and Methods

### 2.1. Study Area

Lima is the capital city of Peru and has over 10 million inhabitants. The city is nestled at 154 m above sea level in the valleys of the Chillón, Rímac, and Lurín rivers, overlooking the Pacific Ocean in the west and the Andes Mountains lying ~3000 m above sea level in the east. The study region spans from ~80 km north to south and 40 km east to west, which includes the city of Lima and the seaport of Callao, together known as the Lima Metropolitan Area.

A grid of 2865 1 $km^2$ pixels was developed to cover the study region, and a 10 km buffer was added to ensure accuracy of any other parameters that need to be interpolated from coarser resolutions down to the modeling grid cells. The added buffer also allowed for better estimation of $PM_{2.5}$ concentrations near the outer boundaries of the study area. With the 10 km buffer, the total number of pixels increased to 5959 during the model development and training period. In Figure 1, we show the study domain and location of ground monitors for the SENAMHI network and Johns Hopkins University (JHU) network as well as the mean $PM_{2.5}$ level at each monitor. The JHU network is part of the Genetic Asthma Susceptibility to Indoor Pollution in Peru, GASP study [32].

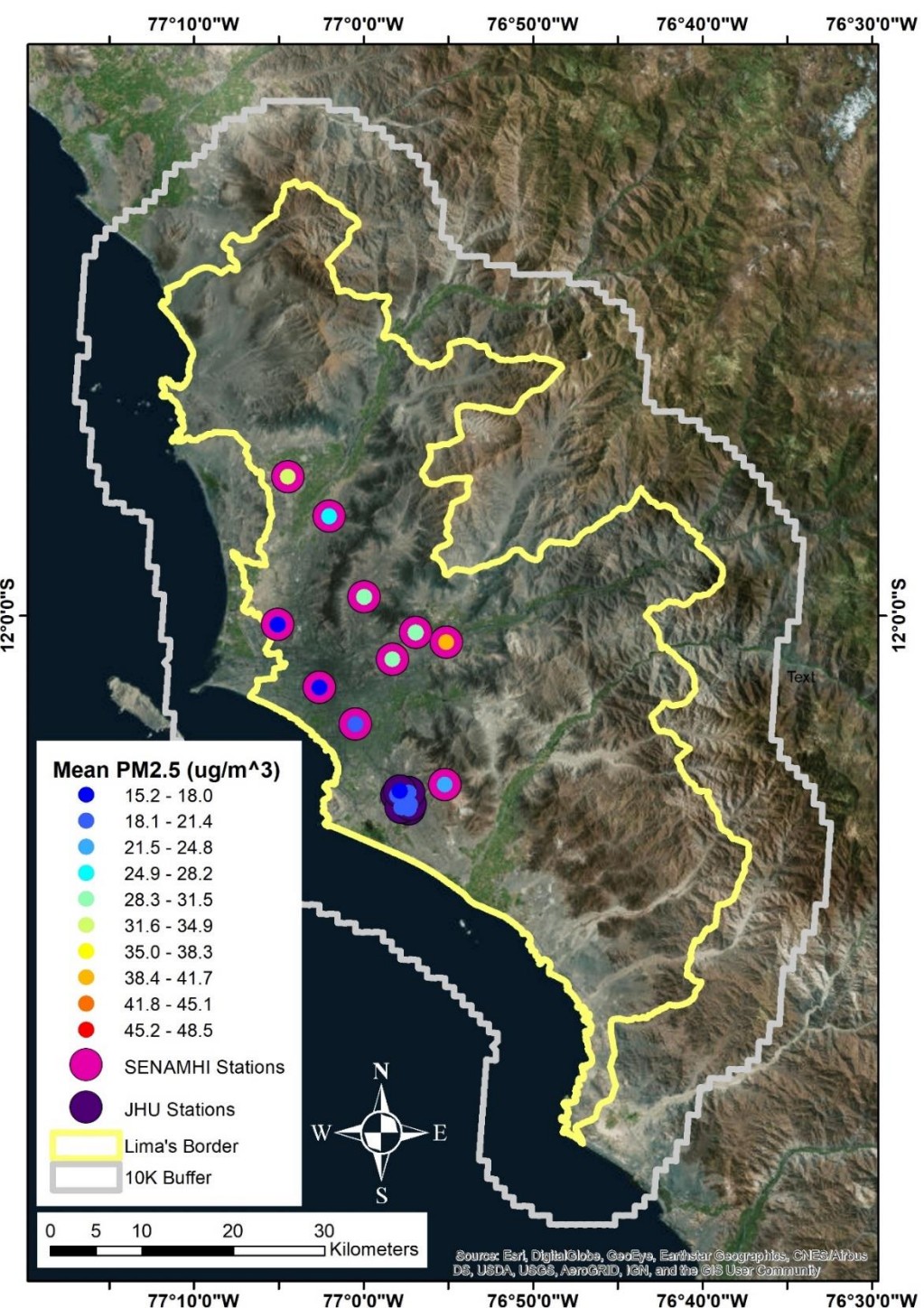

**Figure 1.** Study domain and location of air monitors. The yellow line details the Lima political border while the gray line details the 10 km buffer. The magenta circles denote the location, distribution, and overall mean PM$_{2.5}$ concentrations in µg/m$^3$ of the Servicio Nacional de Meteorología e Hidrología del Perú (SENAMHI) monitor network, while the purple circles denote the same information for the Johns Hopkins University (JHU) monitor network.

## 2.2. Ground PM$_{2.5}$ Data

There are ten SENAMHI stations that measure PM$_{2.5}$ and PM$_{10}$ concentrations in Lima, Peru. These 10 monitoring stations are Thermo Beta 5014i monitors utilizing the beta ray attenuation method and are calibrated three times a year (February, June, and October, starting in October 2014) [33].

SENAMHI stations recorded daily mean measurements of $PM_{10}$ starting in 2010 and $PM_{2.5}$ from 2014 to 2016, and its ten sites contributed 6389 daily observations from 2014 to 2016. Additionally, data from 15 mobile air quality monitors located in Pampas de San Juan de Miraflores were provide by Johns Hopkins University (JHU stations) [34]. These monitors provided one mean estimate each week from November 2011 to March 2013, and were interpolated to the daily level by giving the six preceding days the same concentration as the measured value on the seventh day. One-$km^2$ grids that contained more than 1 JHU station were averaged, which reduced the number of stations from 15 to 6. The JHU sites provided 2081 daily observations from six grid cells to the model fitting dataset. Table 1 shows the elevation and total number of measurements available at each monitor and their respective network.

**Table 1.** $PM_{2.5}$ ground monitor information, elevation, and total number of observations at each monitor and their respective network.

| Network | Station | Elevation (m.) | # of Measurements |
|---------|---------|----------------|-------------------|
| JHU | Station 02 | 94.6 | 339 |
| JHU | Station 07 | 123.6 | 417 |
| JHU | Station 08 | 74.2 | 288 |
| JHU | Station 09 | 186.0 | 443 |
| JHU | Station 10 | 192.1 | 287 |
| JHU | Station 11 | 109.2 | 307 |
| SENAMHI | ATE | 372.7 | 528 |
| SENAMHI | CDM | 124.5 | 544 |
| SENAMHI | CRB | 219.5 | 737 |
| SENAMHI | HCH | 301.2 | 696 |
| SENAMHI | PPD | 186.0 | 778 |
| SENAMHI | SBJ | 131.3 | 581 |
| SENAMHI | SJL | 237.5 | 757 |
| SENAMHI | SMP | 58.5 | 775 |
| SENAMHI | STA | 254.3 | 598 |
| SENAMHI | VMT | 328.3 | 395 |

Note: SENAMHI Station is abbreviated from the name of the location. JHU stations collected measurements from November 2011 to March 2013 and SENAMHI stations collected measurements from April 2014 to December 2016.

*2.3. Satellite Data*

Satellite aerosol optical depth (AOD) at 1 $km^2$ spatial resolution retrieved using the MAIAC (Multiangle Implementation of Atmospheric Correction) algorithm was obtained from the MAIAC science team at NASA's Goddard Space Flight Center. The MAIAC algorithm accomplishes atmospheric correction by first gridding the data to a fixed 1 $km^2$ grid and accumulating of up to 16 days of measurements [35]. Using a time series analysis, the pixels are grouped and the surface bidirectional reflectance distribution function (BRDF) and aerosol parameters over both dark vegetated surfaces and bright surfaces is derived [35].

AOD measurements from Arica [36], the nearest Aerosol Robotic NETwork (AERONET) site located in Chile, were compared to an average of 5 × 5 $km^2$ box of MAIAC AOD centered at the Arica site to assess validity and accuracy from 2010 to 2015. AERONET is a ground-based remote sensing network that provides global observations of AOD [37]. AERONET L2 measurements within 15 min of the MAIAC measurements were used in the validation process to ensure accuracy; however, there may be some uncertainties in the validation results since Arica is located 1017 km northwest of Lima. Nonetheless, AERONET vs. MAIAC AOD validation has been performed in the past showing good agreement [9,38]. The highest annual correlation coefficients between MAIAC AOD and measurements from Arica ranged from 0.59 to 0.74 for Aqua and 0.60 to 0.79 for Terra. The highest correlation coefficient was observed in 2011 for Aqua and 2012 for Terra, with the total number of observations ranging between 42 and 119. Subsequently, an average between Terra and Aqua MAIAC AOD was calculated and gap-filled through a random forest method discussed in Bi et al., which achieved a cross-validation $R^2$ of 0.82 [39]. Daily data for cloud fraction at 5 $km^2$ spatial resolution

was downloaded from the Level-1 and Atmosphere Archive & Distribution System Distributed Active Archive Center (LAADS DAAC) [40] for 2010 to 2016 and processed through IDL. Processes of how cloud fraction data was used in gap-filling MAIAC AOD is described through Bi et al. [39].

## 2.4. Chemical Transport Model (CTM) Data

SENAMHI produces Weather Research and Forecast model coupled with Chemistry (WRF-Chem) simulations for air quality forecasts in Lima at 5 $km^2$ spatial resolution [41]. WRF-Chem is a next-generation atmospheric chemical transport model (CTM) developed by the National Oceanic and Atmospheric Administration (NOAA) and the National Center for Atmospheric Research (NCAR) [42]. CTMs simultaneously simulate the emissions, turbulent mixing, transport, transformation, and fate of trace gasses and aerosols using a combination of meteorological fields, topography data, and emission modules based on measurements of emission factors and ambient concentrations [42]. SENAMHI WRF-Chem configuration has been previously described [41]. In brief, initial meteorological conditions were obtained from the National Centers for Environmental Prediction (NCEP) with emissions inventory derived mainly from anthropogenic vehicular emissions [41]. WRF-Chem data outputs were produced using emissions inventory based on vehicular traffic and packaged in monthly files with 26 vertical layers in the atmosphere every 6 h (00:00, 06:00, 12:00 and 18:00 UTC); however, only the surface layer (vertical layer 0) was used and an average combining all four time measurements were calculated. SENAMHI WRF-Chem parameters used in this study include cloud cover, albedo, surface pressure, temperature, u- and v- wind components, simulated $PM_{2.5}$, and planetary boundary layer height (PBL). There parameters were interpolated to the 1 $km^2$ modeling grid using an inverse distance weighting method.

## 2.5. Meteorological Variables

Data at 6-h increments for 28 parameters including dew point, temperature, wind, and pressure were downloaded for January 2010 through December 2016 from the European Centre for Medium-Range Weather Forecasts (ECMWF) archive [43] at 12.5-$km^2$ spatial resolution [44], and interpolated to the 1 $km^2$ modeling grid using inverse distance weighting. Subsequently, a daily average was calculated for each variable. As part of the cross-validation process, a correlation analysis was performed on temperature, wind, and pressure between WRF-Chem and ECMWF. Furthermore, temperature and dew point from ECMWF was used to calculate relative humidity [45]. In addition, ground meteorological data was downloaded from the Weather Underground website for four individually-owned weather stations along with one airport station. These data were used to evaluate the quality of ECMWF and WRF-Chem meteorological parameters. In Figure S1 of the Supplementary Materials, we show a simple correlation matrix between Weather Underground temperature and relative humidity with WRF-Chem temperature and ECMWF relative humidity to investigate the relationship between measured ground observations and the quality of the forecasted data from ECMWF.

## 2.6. Land Use Variables

Elevation data from the Advanced Spaceborne Thermal Emission and Reflection Radiometer Global Digital Elevation Map (ASTER GDEM) was downloaded from EARTHDATA for Lima, Peru [46]. Census population data for Lima was only available for 2012. To ensure completeness and consistency, LandScan$^{TM}$ yearly population data for 2010 through 2016 was used [47]. Land use parameters at 30-m resolution (open shrubland, bare/sparse vegetation, waterbodies, and artificial/urban areas) for 2010 were derived from the GlobeLand30 product produced by the National High Technology Research and Development Program of China [48]. The 30-m spatial resolution raster was cut into 1 $km^2$ grids to match the MAIAC AOD grid cells, and a percent urbanization was calculated by dividing the area classified as urban in each 1 $km^2$ grid cell by the total area of that cell. Normalized difference vegetation index (NDVI) data at 500 m spatial resolution (MYD13A1 Version 6) was downloaded from the LAADS

DAAC for years 2010 to 2016 [49]. Since NDVI is produced at 16 day intervals, every 15 days preceding the day with measured NDVI was given the same NDVI values. Road Network Data was downloaded as an ArcGIS-ready shapefile from the OpenStreetMap project through Geofabrik [50], and processed in ArcGIS. The road network map was reclassified into three classes—motorways, primary, and trunk roads—and secondary and tertiary roads, and a distance in meters was calculated between the centroid of each study domain grid cell to the nearest segment of road based on class.

## 2.7. Random Forest Model

A random forest (RF) model was used to fit 16 predictors to 8470 ground measurements. The RF model's advantages include its accuracy in learning and classifying features, its ability to include a large number of input variables, and its output of variable importance. Random forest is a supervised machine learning model that works by averaging a set of decision trees that calculates the best predictions based on a subset of predictors [51]. The RF model selects a random subset of samples from all observations with replacement, and subsequently selects the best set of predictors that provides the best split at each node [51]. The two main parameters in a random forest model are the number of predictors sampled for each node ($m_{try}$) of the tree and the number of trees or subset of samples to be averaged ($n_{tree}$). Comparison of results with different settings of $m_{try}$ and $n_{tree}$ was conducted to achieve the best prediction accuracy. The 16 variables used in the random forest model training includes predicted MAIAC AOD from the gap-filling method, NDVI, percent urbanization, road category 3 distance, elevation, population density, interpolated WRF-Chem simulated $PM_{2.5}$, temperature, surface pressure, albedo, cloud fraction, PBL, and wind V and U components, and interpolated relative humidity and surface solar radiation downwards from ECMWF, with $m_{try}$, and $n_{tree}$ set at 6, and 1000, respectively.

A 10-fold cross-validation (CV) process was carried out on the RF model to validate the prediction results. The model fitting dataset, consisting of 8470 ground observations, were randomly divided into 10 segments with each segment containing 10% of the data. Nine of the segments were used as a training dataset set to fit the model and the remaining segment is used as a testing dataset to make predictions. This process was repeated 10 times, each time dividing the dataset at different intervals to ensure that the segments are not repeated. After the 10th repetition, the total number of predictions based on the testing dataset was combined into one dataset and is equal to the original number of ground observations. This CV technique is commonly used in similar studies estimating $PM_{2.5}$ and is better suited for a moderate to small sample size datasets.

## 3. Results

### 3.1. Description of $PM_{2.5}$ Ground-Based Measurements

Daily predictions of $PM_{2.5}$ started on 2 March 2010 and ended on 31 December 2016. In total, 2232 daily predictions were made between 2010 and 2016. In Figure S2 of the Supplementary Materials, we show histograms of all 16 predictors used in the modeling approach. Variables such as MAIAC AOD, surface solar radiation downwards, NDVI, temperature, and PBL were normally distributed. In contrast, variables that are temporally static, such as road distance and elevation, are non-normally distributed.

Figure 2 shows the time series of monthly mean ground measurements at each ground monitor from both the SENAMHI and the JHU networks. Mean (Std. Dev.) $PM_{2.5}$ for all JHU monitors from November 2011 to March 2013 is 18.9 (4.7) $\mu g/m^3$ with mean individual monitors ranging from 16.8 (4.0) (JHU Station 11) to 19.9 (5.8) $\mu g/m^3$ (JHU Station 9). The homogeneity of JHU measurements may be due to the spatial location of these monitors being clustered within 2–3 km in the south region of the study domain. In general, ground JHU measurements peak to 29 $\mu g/m^3$ around April of 2012 and gradually decrease to 12.5 $\mu g/m^3$ in September of 2012 before increasing to a high of 30.5 $\mu g/m^3$ in March of 2013. It is unclear if $PM_{2.5}$ levels peak at this point or continue to increase as data beyond

this period is unavailable for JHU monitors. All JHU monitors share this temporal trend; nonetheless, this similarity may again be due to the clustered location of the JHU monitors.

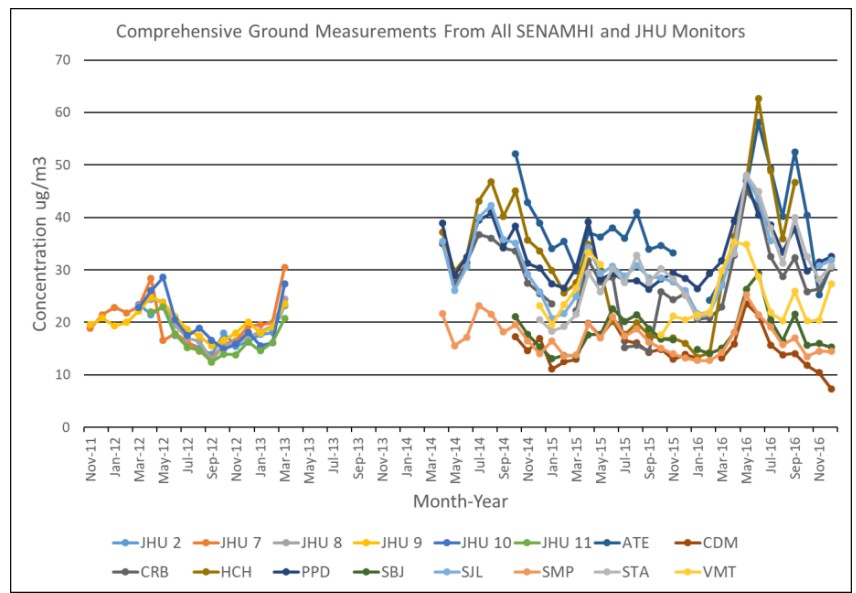

**Figure 2.** Time series of monthly mean ground PM$_{2.5}$ measurements in µg/m$^3$ at each monitor station for both the SENAMHI and JHU networks from November 2011 through December 2016. SENAMHI station names are abbreviated from the name of the location.

SENAMHI measurements show a slightly different temporal pattern. Mean (Std. Dev.) PM$_{2.5}$ for all SENAMHI monitors from April 2014 to December 2016 is 26.7 (11.6) µg/m$^3$ with mean individual monitors ranging from 15.2 (5.3) µg/m$^3$ (Station CDM) to 38.3 (12.2) µg/m$^3$ (Station ATE). SENAMHI PM$_{2.5}$ tend to peak at 52.1 µg/m$^3$ between July and August of 2014 (winter) and gradually decrease to 11.8 µg/m$^3$ around November and December (summer), before increasing again to a peak of 39.2 µg/m$^3$ from March to April of 2015. Temporal trends also indicate PM$_{2.5}$ decreases from May of 2015 to a low of 13.3 µg/m$^3$ in February of 2016 before increasing to a peak of 63.6 µg/m$^3$ in June of 2016. Although most monitors within the SENAMHI network share this temporal trend, there is spatial variation coinciding with the location of the monitors. The three monitors closest to the shore (Stations CDM, SBJ, and SMP) all have the lowest mean PM$_{2.5}$ measurements (15.2, 18.2, and 17.2 µg/m$^3$, respectively), while the three monitors with the highest measurements (ATE: 38.3 µg/m$^3$; PPD: 32.8 µg/m$^3$; and SJL: 31.1 µg/m$^3$) are located further inland closer to the Andes Mountains. The differences in trends between JHU and SENAMHI networks may be a result of the JHU monitors being located in the southern part of Lima, where trends in temperature, winds, and other predictors of PM$_{2.5}$ may be different compared to the SENAMHI stations. Furthermore, SENAMHI stations are distributed across a larger area of the study domain and may have the potential to detect more spatial variability compared to JHU monitors. Although there is variability in the range of PM$_{2.5}$ levels between the two monitoring networks, both networks suggests that PM$_{2.5}$ levels are highest during the Summer; although JHU and SENAMHI stations share peaks in common during the months of March through May, ground measurements are only available for JHU sites from November of 2011 to March of 2013 and from April of 2014 to December of 2015, with no spatial or temporal similarities to the SENAMHI network. Therefore, a continuous and fair comparison of the two networks is not possible.

*3.2. Random Forest Model Performance and Cross-Validation*

A linear mixed effects model (LME) was original conducted (cross-validation (CV) R$^2$ and root mean square error (RMSE) was 0.60 (6.85 µg/m$^3$)); however, the RF model was found to outperform

the traditional LME model. The RF $R^2$ (RMSE) was 0.70 (5.95 µg/m$^3$), and the CV $R^2$ (RMSE) was 0.70 (5.97 µg/m$^3$), indicating that the model is stable and that there is good fit between the predictors and the ground measurements. Figure 3 panel A shows the density plot of CV predicted vs. measured PM$_{2.5}$ concentrations. The slope and intercept from the RF model CV is 1.05 and −1.04 µg/m$^3$, respectively, indicating a good fit (optimal, slope = 1, intercept = 0). Results from our CV indicates that our model slightly overestimate lower PM$_{2.5}$ measurements and underestimates higher PM$_{2.5}$ measurements. Furthermore, in Figure 3 panel B, we show good agreement between the ground measurements and our daily estimate measurements through a Bland–Altman plot. In the Bland–Altman plot, the difference between ground and predicted PM$_{2.5}$ measurements are plotted against the mean of each pair. The mean difference between observations in the CV dataset was −0.09 µg/m$^3$ with a standard deviation of 5.97 µg/m$^3$. The Bland–Altman plot indicates that there is good agreement between the ground and predicted measurements, with 94.5% of the observations falling within 2 standard deviations of the mean differences. Figure 4 shows the importance rankings of each predictor in the RF model, which is a measure of parameter predictive power based on a permutation test. Under the null hypothesis in a random forest model, each predictor variable is not important: the permutation test rearranges the values of that variable to detect any improvement in prediction accuracy [51,52]. The RF model suggests that surface downward solar radiation, temperature, relative humidity, PBL, and AOD are the most important predictors of PM$_{2.5}$.

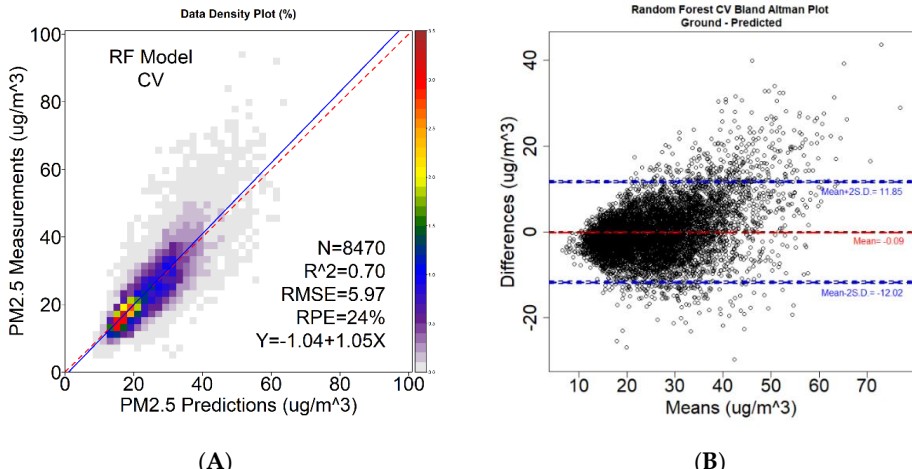

(**A**)          (**B**)

**Figure 3.** (**A**) Density plot of ground and predicted PM$_{2.5}$ measurements in µg/m$^3$ based on the cross-validation of the Random Forest model. (**B**) Bland–Altman plot of differences between ground and predicted PM$_{2.5}$ in µg/m$^3$ against the means of each pair. This plot shows good agreement as 94.5% of observation pairs fall within 2 standard deviations of the mean difference.

Figure 5 shows a time series of monthly mean ground measurements and predictions from the RF model for each ground monitor. The RF model is able to track well the temporal variability of the ground monitors, but tends to underestimate higher peaks and overestimate the low points. This trend is observed in both the SENAMHI and JHU networks. We show the predicted annual mean PM$_{2.5}$ concentrations across our study region in µg/m$^3$ in Figure 6. Mean annual PM$_{2.5}$ concentrations start at 14.6 µg/m$^3$ along the coastline and gradually increases up to 48.5 µg/m$^3$ against the Andes Mountains on the east. Monitors with lower mean PM$_{2.5}$ measurements are also those that are located closer to the coastline, and are at a lower elevation. Temporally, PM$_{2.5}$ levels are highest during 2010 and dip during 2011 to 2014, before increasing back up in 2015 through 2016. Although ground measurements are not available for 2010, the increase in predicted mean annual PM$_{2.5}$ from 2015 to 2016 can be observed in the monthly mean measurements from the SENAMHI monitors (Figure 2), which show a spike in PM$_{2.5}$ during the months of April and May of 2016 compared to relatively lower levels in 2015. Month-to-month variation can be seen in Supplementary Figure S3. PM$_{2.5}$ is highest starting

from April through October (highest in May–June, winter) before decreasing during the months of November to March (lowest in February, summer). Although this monthly trend is different from those observed in the JHU ground measurements, they are consistent with monthly mean SENAMHI ground measurements. This may be due to a smaller number of ground measurements for JHU compared to SENAMHI in the model fitting dataset. Furthermore, JHU monitors produced weekly measurements, which had to be interpolated to daily estimates for model fitting; therefore, monthly trends may not be meaningful for JHU measurements.

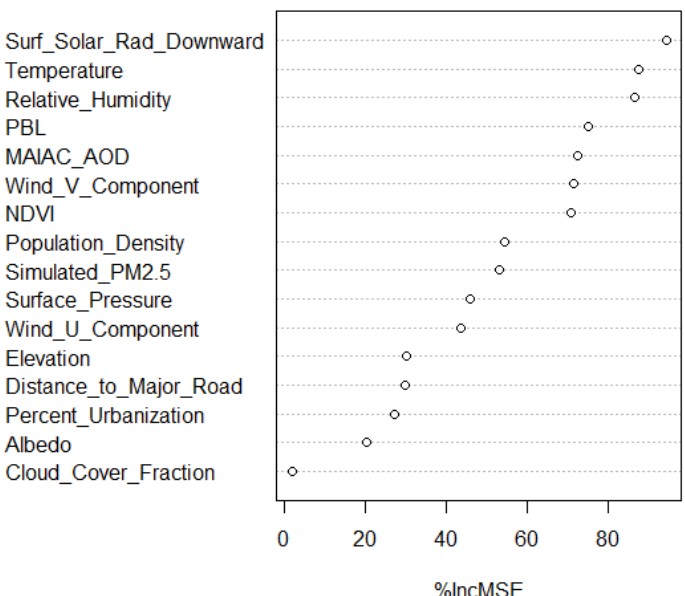

**Figure 4.** Importance of each variable in the random forest model by percent increase mean square prediction error (MSE).

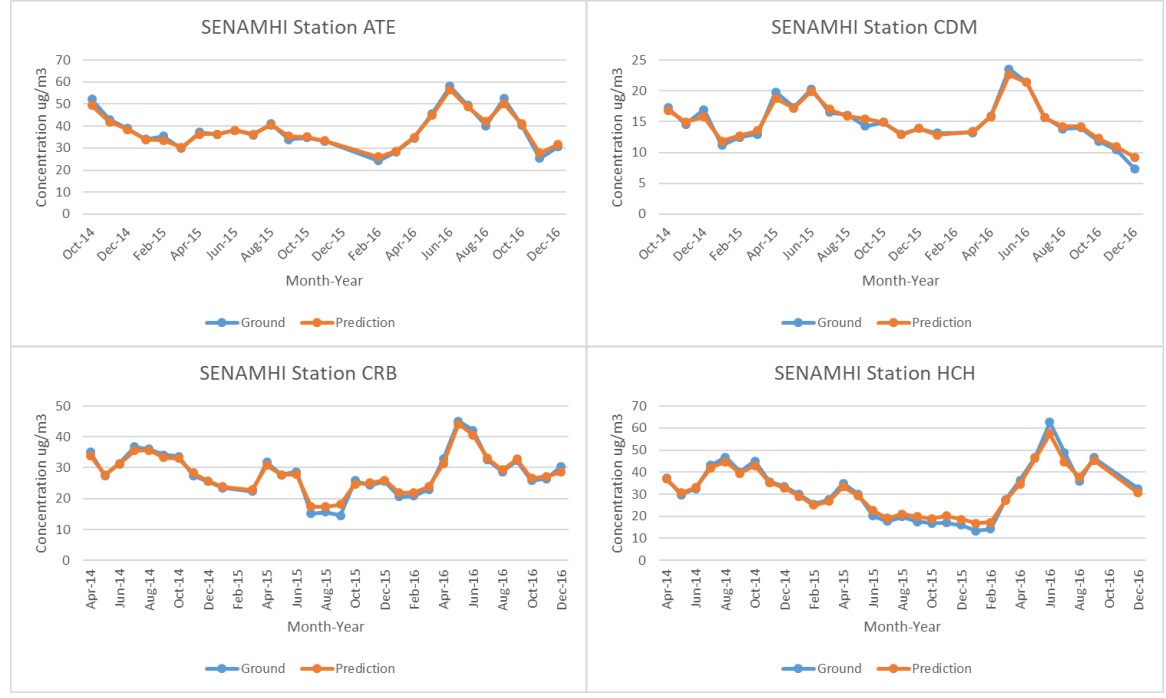

**Figure 5.** *Cont.*

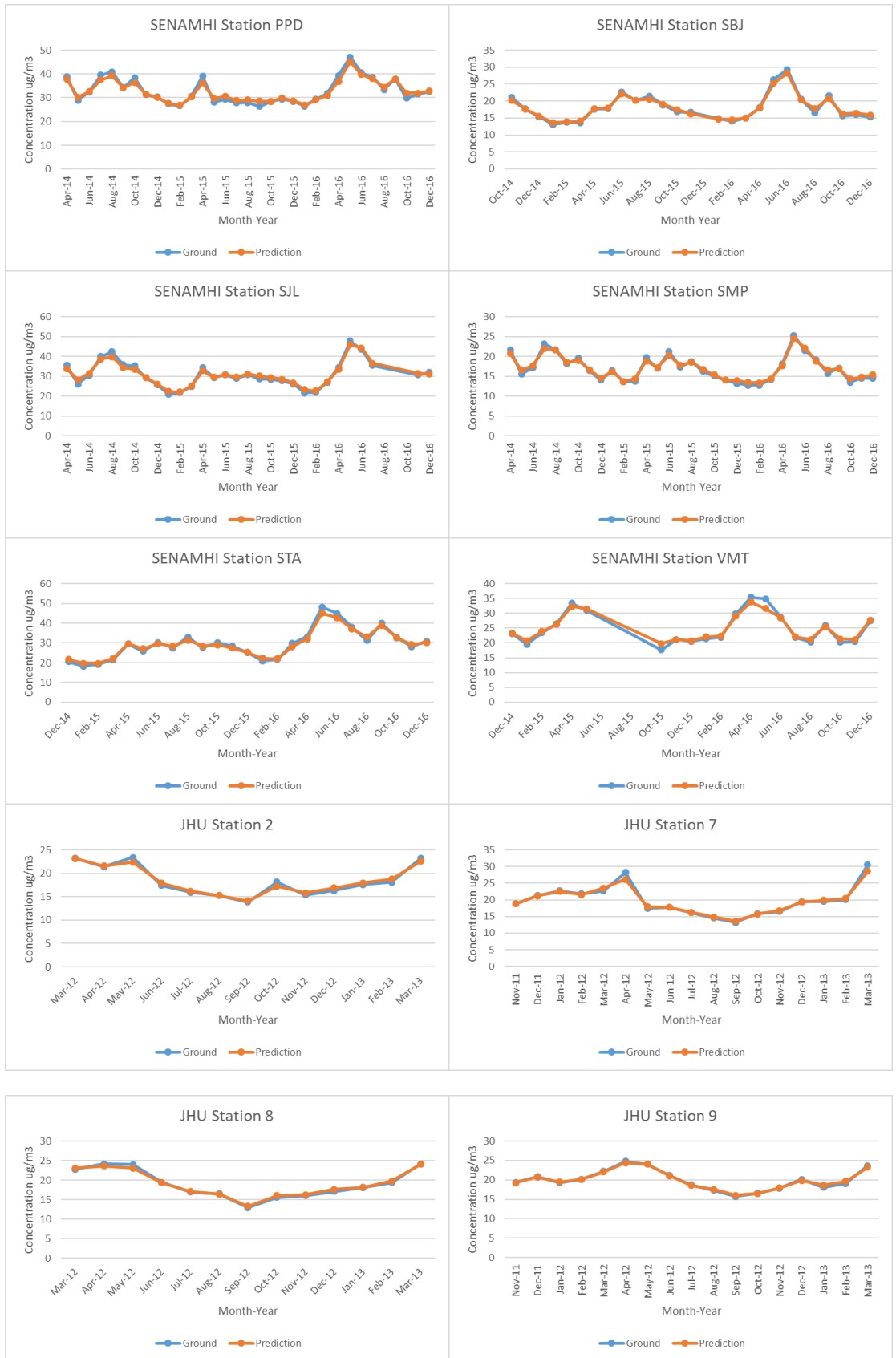

**Figure 5.** *Cont.*

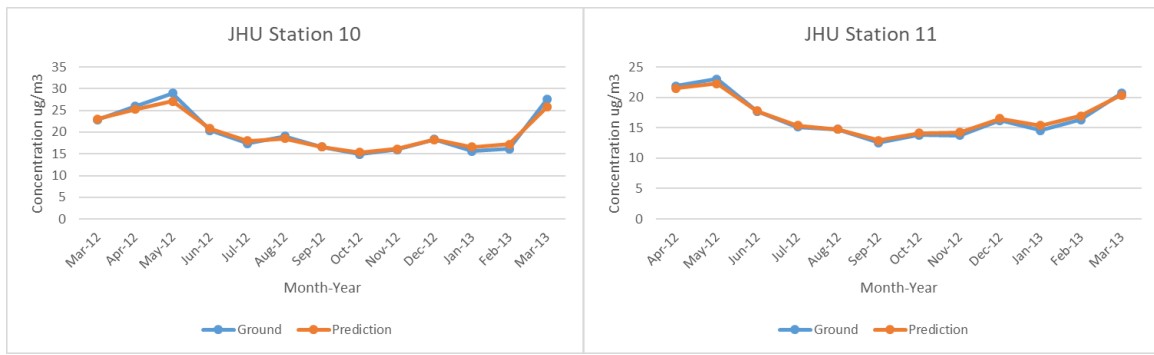

**Figure 5.** Time series of monthly mean ground measurements and predicted PM$_{2.5}$ in μg/m$^3$ based on random forest model at each monitor station. SENAMHI station names are abbreviated from the name of the location.

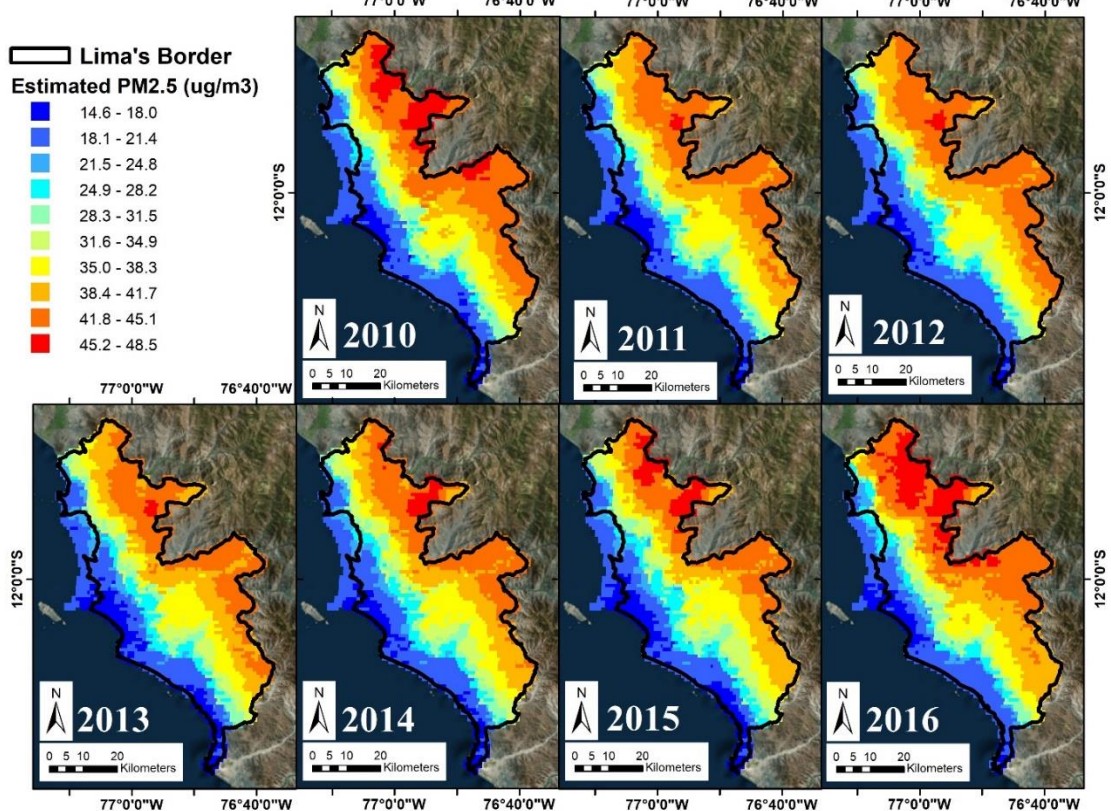

**Figure 6.** Annual mean prediction maps of PM$_{2.5}$ in μg/m$^3$ from the random forest model in Lima, Peru from 2010 to 2016.

## 4. Discussion

Until recently, studies modeling the concentration of PM$_{2.5}$ have been limited in South America due to lack of ground monitoring data. Previous studies have estimated historical ambient PM$_{2.5}$ concentrations globally from a combination of satellite remote sensing data and chemical transport models; however, these studies were conducted at coarse resolution (e.g., $10 \times 10$ km$^2$) and were evaluated by ground PM$_{2.5}$ measurements from the literature. Furthermore, results from these studies do not provide daily measurements to aid in epidemiological health studies [53]. Brazil, Chile, Colombia, Ecuador, and Peru are the few countries with existing PM$_{2.5}$ monitors in South America prior to February of 2016; yet, Chile is the only country with known spatiotemporal and forecast models of PM$_{2.5}$ [54,55]. The Chilean PM$_{2.5}$ model was constructed using three winter months of

hourly $PM_{2.5}$ measurements from 11 monitors and incorporated CTMs; however, their model did not incorporate satellite remote sensing techniques to enhance prediction capabilities, and could only forecast $PM_{2.5}$ levels in the proceeding 48-h period [54]. The only current existing model of $PM_{2.5}$ in Peru is constructed through kriging techniques using ArcGIS for the province of Cusco [56]. The Cusco model was derived from a singular fixed monitor that recorded 24-h time-integrated samples for only 12 days during July 2005, and measured $PM_{2.5}$ at "subjectively chosen hot spots" using standalone laser photometers to augment ground measurements [56]. Although this study may provide support for short-term acute exposure of $PM_{2.5}$ health studies, it does not provide daily historical measurements for epidemiologic studies that investigate population health effects due to acute exposure to $PM_{2.5}$, especially outside of Cusco, like Lima, where pollution levels are much higher.

Our $PM_{2.5}$ model is the first advanced model in Peru to incorporate both satellite remote sensing data and CTM outputs to provide daily ground measurements at 1-$km^2$ resolution in Lima, the most populated and polluted region of Peru, to aid in epidemiologic studies. A major strength of this study is the ability to estimate $PM_{2.5}$ in Lima at a high resolution through the implementation of MAIAC gap-filled AOD. Our finer-scale model is able to capture local spatiotemporal trends and, compared to coarser resolution products, are better suited for use in epidemiological health studies that require daily measurements of exposure at fine-resolution. Additionally, predictions from our model correspond well at each ground monitor station (as seen in Figure 5). Maximum concentrations are typically observed between May and September (winter months), with minimum concentrations generally observed between October and April (summer months); however, these trends vary from year to year and between each monitoring site. Furthermore, monthly variation in $PM_{2.5}$ concentrations is also affected by meteorological conditions present in Lima. In the summer months, Lima is subjected to smaller and less permanent marine thermal inversion due to the Humboldt oceanic current in the west. The result is a decrease in stratiform clouds and an increase in solar irradiation in conjunction with lower relative humidity and higher temperatures, which leads to resuspension of course PM and the prevention of secondary PM formation, decreasing the levels of $PM_{2.5}$ [18]. However, during winter there is an increase in stratiform clouds along with an increase in relative humidity and light precipitation, resulting in wet deposition of $PM_{10}$ and a subsequent increase in $PM_{2.5}$ due to secondary formation via converted gas particulate [18].

Nonetheless, our study uses an emerging ensemble classifier—the random forest model—to generate our estimates which comes with limitations and uncertainties. Currently, annual predictions from the RF model show that concentrations of $PM_{2.5}$ are lowest near the coast, and in and around the urban centers of Lima, while gradually rising with elevation up to the Andes Mountains. This may be driven by the fact that all ground $PM_{2.5}$ monitors are located below 500 m above sea level, and monitors located at lower elevation have lower $PM_{2.5}$ levels. As a result, when $PM_{2.5}$ levels are extrapolated beyond the existing ground data, their levels continue to increase with elevation up to the mountains and predictions made at elevation above 1000 m may contain more uncertainty. Furthermore, the average height of JHU monitors is located at 132.7 (Std. Dev. = 43.6) meters above ground, while the mean height for SENAMHI monitors is 213.4 (Std. Dev. = 90.9) meters, indicating that SENAMHI monitors have a wider range of elevation height compared to JHU monitors. Additionally, JHU monitors also have a more homogeneous level of $PM_{2.5}$ since their daily values were interpolated from weekly measurements and comprised of 25% of the total ground measurements, which may add to the explanation of why elevation had relatively lower importance in the RF model. To counter the effects of elevation in the model, distance from shoreline was added to the model as a predictor. Although distance from coast should have explained much of the variation in $PM_{2.5}$ as the annual maps suggests, this variable did not improve the "out of bag" $R^2$ in the RF model and also did not change the resulting predictions maps and was subsequently discarded from the final model. A possible reason for why distance from coast did not improve model performance may be due to the cluster of JHU monitors all residing close to the coast. Because of their proximity to each other, as well as to the coast, the JHU monitors do not exhibit enough spatial variability both in terms of $PM_{2.5}$ levels to impact

model performance. Furthermore, Lima's distinct topography and geographic location also lends to the spatial distribution of $PM_{2.5}$ concentrations. As discussed previously, much of Lima's production of $PM_{2.5}$ stems from an aging vehicular fleet located mostly in the densely populated urban areas in and around the metropolitan cities. Additionally, $PM_{2.5}$ is also being produced in rural areas from biomass burning as fuel. The spatial pattern of $PM_{2.5}$ seen in the annual prediction maps may be a result of persistent and prevailing coastal winds from the south and southwest pushing pollutants from the coastal cities and trapping them against the Andes Mountains in the east and northeast [18]. This phenomenon is similar to that seen in the Los Angeles Basin, where the topography is nearly identical to that of Lima with prevailing coastal winds blowing pollutants against the Transverse Ranges [57]. Nonetheless, census data indicate that the number of residents living above 1000 m above sea level is relative small and may not impact future epidemiologic studies.

Consequently, a limitation of this study is the lack of monitors located at higher altitudes to validate our results. All monitors are located centrally in the urbanized metropolitan area of Lima, with no monitors in the far corners of the North, East, and South in our study domain. Furthermore, all JHU monitors are clustered within a few kilometers of each other in the mid-southern region of Lima, covering six of the 2970 grid cells in the study domain, which may affect their predictive capabilities on the rest of the study domain leading to the lack of spatial variability from north to south in the study domain. Additionally, JHU ground measurements were collected from late 2011 to early 2013, while the SENAMHI measurements were collected from mid-2014 through 2016, which impact model predictive abilities across the years (i.e., borrowing prediction capabilities of JHU measurements to estimate $PM_{2.5}$ in the entire study domain for 2014 to 2016 and conversely borrowing prediction capabilities of SENAMHI measurements to estimate $PM_{2.5}$ in the entire study domain from 2011 to 2013). Nonetheless, JHU measurements served the purpose of increasing our sample size and helped make our model more stable and robust. When JHU measurements were not included in our RF model (the CV R2 was 0.67 (RMSE = 6.68 $\mu g/m^3$)), they were subsequently kept in the model fitting dataset to enhance not only sample size, but also to provide additional spatial and temporal quality to the ground measurements. Finally, before utilizing and applying the model-derived dataset in epidemiological studies, future research will focus on evaluating model forecasting capacity on a daily basis. Furthermore, the SENAMHI ground monitors have longer periods of $PM_{10}$ measurements. Future study will also explore converting $PM_{10}$ measurements to $PM_{2.5}$ to maximize ground observations in the model fitting process [58]. Silva et al. studied the relationship between $PM_{2.5}$ and $PM_{10}$ concentrations at each of the 10 SENAMHI stations with Pearson correlation coefficients ranging from 0.49 to 0.72, and that the annual $PM_{2.5}/PM_{10}$ for the stations range from 0.21 to 0.44, indicating that $PM_{2.5}$ concentrations represent 21% to 44% of the total $PM_{10}$ in Lima [18].

## 5. Conclusions

Our satellite-driven $PM_{2.5}$ exposure model is the first of its kind in both Lima and South America, incorporating satellite remote sensing data, meteorological fields from chemical transport models, and land use parameters to estimate daily $PM_{2.5}$ measurements at 1-km resolution, with greater spatial and temporal coverage than previous studies conducted in Peru. Predicted daily $PM_{2.5}$ levels by our model allow for construction of consistent long-term historical measurements that bridges the data gaps created by sparse data quality from both the SENAMHI and JHU monitor networks, and would provide strong data support for epidemiologic studies that focus on both cardiovascular and respiratory outcomes in Lima. Our future research will focus on converting $PM_{10}$ to $PM_{2.5}$ from the SENAMHI monitors to maximize ground observations across years prior to 2014, and improve model stability and precision, and further improve on the accuracy of our predictions for use in urgently needed epidemiologic studies to assess the impact of air pollution in Lima, Peru.

**Supplementary Materials:** The following are available online at http://www.mdpi.com/2072-4292/11/6/641/s1, Figure S1: Simple Correlation Matrix between Weather Underground temperature and relative humidity with WRF-Chem temperature and ECMWF relative humidity, Figure S2: Histograms of each predictor variable, Figure S3: Monthly mean prediction maps of $PM_{2.5}$ concentrations in µg/m$^3$ for 2015.

**Author Contributions:** Conceptualization, Y.L., K.S. and G.F.G.; Methodology, Y.L. and B.N.V.; Validation, B.N.V.; Formal Analysis, B.N.V.; Data Curation, B.N.V, K.S., O.S., N.N.H, and W.C.; Writing–Original Draft Preparation, B.N.V.; Writing–Review & Editing, B.N.V., Y.L, K.S., N.N.H, W.C. and G.F.G.; Visualization, B.N.V..; Supervision, Y.L., K.S., and G.F.G.; Resources, J.B. and Q.X.; Funding Acquisition, K.S. and Y.L.

**Funding:** Research reported in this publication was supported by the NIH Fogarty International Center, National Institutes of Environmental Health Sciences (NIEHS) R01ES018845, R01ES018845-S1, National Cancer Institute, National Institute for Occupational Safety and Health, and the NIH under Award Number U01 TW0101 07. The content is solely the responsibility of the authors and does not necessarily represent the official views of the National Institutes of Health. This research was also support by the HERCULES Center Pilot Project Program. We also express great thanks to Gustavo Gonzales for his support in this project.

**Conflicts of Interest:** The authors declare no conflict of interest.

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
