# Peer review of "Developing an Advanced PM2.5 Exposure Model in Lima, Peru"

_remotesensing, doi:10.3390/rs11060641_

Round 1

Reviewer 1 Report

This work is important because it is the first of its kind to be carried out in the city of Lima. However, the authors exhibited the results without deep insight into the scientific analysis. The presentation of some figures needs to be improved. Introduction should provide a state of the art regarding PM modelling and why random forest was applied in this case, citing other previous works using this methodology.

Comments and suggestions:

·      Section 1: Bibliography about previous air pollution studies in South America is required

·      Section 2.2: Can you tell something about equipment calibration? If they follow any international standards ? the error associated to measurements ?

·      Section 2.3:

I think that validation of satellite data needs to be in another section. It would be more appropriate for this section to be in results. Or, if you consider that should be when satellite data is presented, at least it should be in a separate paragraph.

You need to describe what AERONET stands for and describe a little what AERONET is.

Uncertainties in validation related to the distance between Arica and Lima (1200 km!) should be mentioned.

Moreover, there are previous works which validated MAIAC data in Southamerica that are not mentioned in the text. I suggest to cite them:

Della Ceca et al 2018 https://doi.org/10.1016/j.isprsjprs.2018.08.016

Martins et al. 2018 https://doi.org/10.1016/j.isprsjprs. 2018.05.013.

·      Section 3.1:

* Title of this section ‘Descriptive Statistics’ should change. Suggestions: ‘PM2.5 ground-based measurements description’ or similar

*Table 1: This table should be in ‘Data and Method’s section, where PM stations are described (section 2.2, without mean concentration values which are already described in the text in section 3.1)

* Standard deviation of mean valued should be reported.

*How can you explain the temporal pattern observed in SENAMHI and JHU measurements? Did any particular events occur during high PM periods? Is important to mention them. Biomass burning is an important factor affecting PM trends in SouthAmerica as is well documented in bibliography.

*It would be nice to see in this section a map (similar to that in Figure 1) with the PM stations marker with a color scale indicating the PM mean for each station.

* Improve Figure 2 please. Enlarge text in figure, preferably in black (not gray), lines on the axes.

·  Section 3.2:

* Lines 298 to 301: ‘The RF model suggests that surface downward solar radiation, temperature, relative humidity, PBL and AOD are the most important predictors of PM2.5, which is logical since these meteorological variables are directly involved in the secondary production of PM2.5. ‘

I do not agree. AOD is not a meteorological variable. PBL is not directly involved in the secondary production of PM2.5, but in the aerosol distribution in the air column. Could you cite bibliography supporting this statement?

Figure 4 also shows that NDVI and wind v component have a similar importance to MAIAC AOD in the Random Forest model. How can you explain this?

*  Figure 6: It would be interesting to add also in this map the PM stations marker with a color scale indicating the PM mean for each station for direct comparison to the model.

·  Section 4:

* Discussion should be improved. Uncertainties and limitations of the model should be better discuss.

* Lines 345-349: ‘Although studies exist that estimate the long-term average of ambient PM2.5 concentrations globally using a combination of satellite remote sensing data and data from chemical transport models, they were conducted at coarse resolution (e.g., 10 x 10 km2), were evaluated against ground PM2.5 measurements from the literature, and do not provide daily  measurements to aid in epidemiological health studies [31].’

Please rewrite this statement. Is confusing.

Minor issues:

·      Line 34 and 33: ‘mean annual maps show’ (not shows)

·      Line 42 ‘of sources including industrial sources, power generation, engine combustion, biomass burning, and’

Not repeat ‘sources’ (suggestion: industrial sources -> industry)

·      Line 48: Please cite some of those air pollution studies in South America (there are not few). Suggestions:

·        Della Ceca et al 2018 https://doi.org/10.1016/j.isprsjprs.2018.08.016

·        Amarillo et al 2012 https://doi.org/10.1016/j.envpol.2012.07.005

·        de miranda et al 2012 https://doi.org/10.1007/s11869-010-0124-1

·      Line 60: ‘The rise in air pollution …’

To what rise in air pollution do you refer to?, there is no increase described in previous sentences.

·      Line 63: Please cite some of those publications related to the ‘limited number of studies on the association between ambient air pollution and health risks in Lima.’

·      Line 64: idem comment in Line 60

·      Line 66 and 67: ‘Current studies on the association between health outcomes and air pollution in Lima have been cross-sectional in design with childhood asthma as the outcome [9, 10].

Rewrite sentence please.

·      Line 119: ‘The study region was divided into 2,970 1 km2 pixels…’

This is part of the description od the data included in the model. I suggest at least to start a new paragraph: ‘A grid of  .. pixels was developed covering the area study in order to…’

·      Line 124: Describe what SENAHAMI stands for when first appear in text

·      Figure 1 caption: JHU: stands for…?  In Line 125: add Johns Hopkins University (JHU)

·      Line 124 and 125:  Figure 1 does not show ‘data from mobile air quality monitors collected and provided by..’ it shows only the stations location …

·      Please, include the web page that the authors include in the text, put as references. For example in Lines 160 – 161 180, 186, 194, 195)

·      Line 154 : ‘for 2010 to 2015’ should be ‘from 2010…’

·      Line 166 and other: Please, first you should mention the Institution and in brackets its acronym. For example: ‘WRF-Chem (Weather Research and Forecast model coupled with 164 Chemistry)’ change for ‘Weather Research and Forecast model coupled with Chemistry (WRF-Chem)’

·      Line 202: NDVI stands for? For vegetation cover we consider the Normalized …

·      Line 203: Please mention the NDVI product code you use.

·      Line 215: suggestion: ‘The two main parameters in a random forest model are the number of predictors sampled for each node (mtry) of the tree and the number of trees or subset of samples to be averaged (ntree).’

·      Line 278: ‘Therefore, a continuous and fair comparison of the two networks are not possible.’

Change are for is (you are talking about the comparison)

Author Response

Comments and Suggestions for Authors

This work is important because it is the first of its kind to be carried out in the city of Lima. However, the authors exhibited the results without deep insight into the scientific analysis. The presentation of some figures needs to be improved. Introduction should provide a state of the art regarding PM modelling and why random forest was applied in this case, citing other previous works using this methodology.

Comments and suggestions:

Section 1: Bibliography about previous air pollution studies in South America is required

Response: Reference 8-14 added to knowledge previous air pollution studies in South America.

Section 2.2: Can you tell something about equipment calibration? If they follow any international standards ? the error associated to measurements ?

Response: Information about equipment, calibration, and method used for collecting PM2.5 samples added in section 2.2. Furthermore, SENAMHI calibrates the 1405 TEOM™ Continuous Ambient Particulate Monitor (this monitor measures PM10) four times a year. In addition to this, SENAMHI calibrates the Model 5014i Continuous Ambient Particulate Monitor continuously (measures the mass concentration of PM2.5) every three months.

Section 2.3:

I think that validation of satellite data needs to be in another section. It would be more appropriate for this section to be in results. Or, if you consider that should be when satellite data is presented, at least it should be in a separate paragraph.

Response: Validation of MAIAC AOD with AERONET separated into a new paragraph.

You need to describe what AERONET stands for and describe a little what AERONET is.

Response: Information describing AERONET and Arica is added to section 2.3.

Uncertainties in validation related to the distance between Arica and Lima (1200 km!) should be mentioned.

Response: Addition of a sentence that mentions the uncertainties in validation Arica AOD.

Moreover, there are previous works which validated MAIAC data in Southamerica that are not mentioned in the text. I suggest to cite them:

Della Ceca et al 2018 https://doi.org/10.1016/j.isprsjprs.2018.08.016

Martins et al. 2018 https://doi.org/10.1016/j.isprsjprs. 2018.05.013.

Revision: These studies are now cited in section 2.3.

Section 3.1:

Title of this section ‘Descriptive Statistics’ should change. Suggestions: ‘PM2.5 ground-based measurements description’ or similar

Response: Title of section is now “Description of PM2.5 Ground-Based Measurements”.

*Table 1: This table should be in ‘Data and Method’s section, where PM stations are described (section 2.2, without mean concentration values which are already described in the text in section 3.1)

Response: Table has been moved to section 2.2. Mean concentrations have been removed, instead mean concentrations can be seen visually in figure 1.

Standard deviation of mean valued should be reported.

Response: standard deviation of mean values added in text.

*How can you explain the temporal pattern observed in SENAMHI and JHU measurements? Did any particular events occur during high PM periods? Is important to mention them. Biomass burning is an important factor affecting PM trends in SouthAmerica as is well documented in bibliography.

Response: the temporal patterns of PM2.5 observed in SENAMHI and JHU measurements could be impacted by different meteorological patterns in Lima. This explanation is reported in the discussion. There were no particular events occurring during high PM periods that we know of.

*It would be nice to see in this section a map (similar to that in Figure 1) with the PM stations marker with a color scale indicating the PM mean for each station.

Response: mean PM for each station is incorporated into Figure 1.

Improve Figure 2 please. Enlarge text in figure, preferably in black (not gray), lines on the axes.

Response: Figure 2 has been revised to show larger text with black instead of gray lines.

Section 3.2:

Lines 298 to 301: ‘The RF model suggests that surface downward solar radiation, temperature, relative humidity, PBL and AOD are the most important predictors of PM2.5, which is logical since these meteorological variables are directly involved in the secondary production of PM2.5. ‘

I do not agree. AOD is not a meteorological variable. PBL is not directly involved in the secondary production of PM2.5, but in the aerosol distribution in the air column. Could you cite bibliography supporting this statement?

Response: the second portion of that statement has been removed.

Figure 4 also shows that NDVI and wind v component have a similar importance to MAIAC AOD in the Random Forest model. How can you explain this?

Response: as a commonly reported limitation, despite the reported variable importance ranks, the random forest model is still very much a black box machine learning method. The importance ranks simply indicate that these two variables contribute relatively equally in making accurate PM2.5 estimates in our study domain. This finding may not be generalizable to other regions or study periods. 

Figure 6: It would be interesting to add also in this map the PM stations marker with a color scale indicating the PM mean for each station for direct comparison to the model.

Response: Our monthly time-series in figure 5 depicts the direct comparison between monitoring station and prediction, which should be more informative and precise as compared to an annual mean comparison.

Section 4:

Discussion should be improved. Uncertainties and limitations of the model should be better discuss.

Response: Impact of meteorology and explanation for elevation and distance to coastline importance in added in the discussion.

Lines 345-349: ‘Although studies exist that estimate the long-term average of ambient PM2.5 concentrations globally using a combination of satellite remote sensing data and data from chemical transport models, they were conducted at coarse resolution (e.g., 10 x 10 km2), were evaluated against ground PM2.5 measurements from the literature, and do not provide daily  measurements to aid in epidemiological health studies [31].’

Please rewrite this statement. Is confusing.

Response: This statement has been revised to “Previous studies have estimated historical ambient PM2.5 concentrations globally from a combination of satellite remote sensing data and chemical transport models; however these studies were conducted at coarse resolution (e.g., 10 x 10 km2) and were evaluated by ground PM2.5 measurements from the literature. Furthermore, results from these studies do not provide daily measurements to aid in epidemiological health studies [53].”.

Minor issues:

Line 34 and 33: ‘mean annual maps show’ (not shows)

Revision: “shows” has been changed to “show”.

Line 42 ‘of sources including industrial sources, power generation, engine combustion, biomass burning, and’

Not repeat ‘sources’ (suggestion: industrial sources -> industry)

Response: “industrial sources” has been changed to “industry”.

Line 48: Please cite some of those air pollution studies in South America (there are not few). Suggestions:

Della Ceca et al 2018 https://doi.org/10.1016/j.isprsjprs.2018.08.016

Amarillo et al 2012 https://doi.org/10.1016/j.envpol.2012.07.005

de miranda et al 2012 https://doi.org/10.1007/s11869-010-0124-1

Response: These studies have been cited.

Line 60: ‘The rise in air pollution …’

To what rise in air pollution do you refer to?, there is no increase described in previous sentences.

Response: The sentence refers to how biomass burning amplifies the levels of PM2.5 in conjunction with vehicular emission. The statement has been revised.

Line 63: Please cite some of those publications related to the ‘limited number of studies on the association between ambient air pollution and health risks in Lima.’

Response: References have been added.

Line 64: idem comment in Line 60

Response: The sentence has been reworded to clarify the meaning of increase.

Line 66 and 67: ‘Current studies on the association between health outcomes and air pollution in Lima have been cross-sectional in design with childhood asthma as the outcome [9, 10].

Rewrite sentence please.

Response: Sentence has been reworded to “Many of the studies investigating air pollution in Lima have been cross-sectional in design, with childhood asthma as a popular health outcome [19,20].“.

Line 119: ‘The study region was divided into 2,970 1 km2 pixels…’

This is part of the description od the data included in the model. I suggest at least to start a new paragraph: ‘A grid of  .. pixels was developed covering the area study in order to…’

Response: The sentence has been reworded for clarity.

Line 124: Describe what SENAHAMI stands for when first appear in text

Response: SENAMHI has already been described in line 110.

Figure 1 caption: JHU: stands for…?  In Line 125: add Johns Hopkins University (JHU)

Response: Description of JHU was added to figure 1 caption. (JHU) was added to line 125.

Line 124 and 125:  Figure 1 does not show ‘data from mobile air quality monitors collected and provided by..’ it shows only the stations location …

Response: This sentence was reworded to “In Figure 1, we show the study domain and location of ground monitors for the SENAMHI network and Johns Hopkins University (JHU) network as well as the mean PM2.5 level at each monitor.”.

Please, include the web page that the authors include in the text, put as references. For example in Lines 160 – 161 180, 186, 194, 195)

Response: References have been added for any webpage listed in the main text.

Line 154 : ‘for 2010 to 2015’ should be ‘from 2010…’

Response: “For” has been changed to “From”.

Line 166 and other: Please, first you should mention the Institution and in brackets its acronym. For example: ‘WRF-Chem (Weather Research and Forecast model coupled with 164 Chemistry)’ change for ‘Weather Research and Forecast model coupled with Chemistry (WRF-Chem)’

Response: Acronyms have brackets have been moved to after the description.

Line 202: NDVI stands for? For vegetation cover we consider the Normalized …

Response: NDVI is now spelled out in text.

Line 203: Please mention the NDVI product code you use.

Response: NDVI product code is added in text.

Line 215: suggestion: ‘The two main parameters in a random forest model are the number of predictors sampled for each node (mtry) of the tree and the number of trees or subset of samples to be averaged (ntree).’

Response: This sentence has been reword per the reviewer’s suggestion.

Line 278: ‘Therefore, a continuous and fair comparison of the two networks are not possible.’

Change are for is (you are talking about the comparison)

Response: “Are” has been changed to “is”.

Reviewer 2 Report

The manuscript titled “Developing an Advanced PM2.5 Exposure Model in Lima, Peru”, by Vu et al., presents a PM2.5 exposure model that incorporates satellite remote sensing, meteorological and land use data to estimate daily PM2.5 measurements at 1 km resolution. The authors claim that the model is the first of its kind in South America, providing greater spatial and temporal coverage than previous studies conducted in Peru. The output of this model will reduce the sparsity of PM2.5 measurements and could be applied to support future epidemiological studies.

The work presents several weakness that must be solved before publication. Firstly, the variable importance shown in figure 4 is very strange. Looking at figure 6, the spatial distribution of PM2.5 in the study area, it is obvious that “elevation” and “distance to shoreline” are very relevant parameters of the distribution. It is worrisome that the model doesn´t capture it. How do the authors explain that the distance from shoreline did not contribute (mentioned in lines 380-384), as it is fairly obvious from the figures. Better analysis must be provided to discard a model problem.

Secondly, the validation of the AOD values with well-established data from the AERONET network (Section 2.3 Satellite data) must be expanded and the details of the comparison clearly explained. It is worrisome that the only reference is from a congress abstract without peer-review. (It can be located at https://ehp.niehs.nih.gov/doi/10.1289/isesisee.2018.O03.03.05 and it mentions: “This abstract was presented at the ISES-ISEE 2018 Joint Annual Meeting and has not been peer reviewed”).

Finally, the future research must be reconsider, The authors express their intention to convert PM10 data into PM2.5 (lines 404-405). PM10 measurement can follow very different trends than PM2.5. depending on the size distribution of the aerosols, and it can lead to wrong conclusions. Also, before applying the model-derived dataset to epidemiological studies (line 413), the model must prove its forecasting capacity on daily basis. Such study was not attempted in this work and it should be the step ahead.

Adding to these, some parts of the manuscript needed clarification, included as notes in the attached file. Taking all this into account, my recommendation is that a major revision is necessary.

Author Response

Comments and Suggestions for Authors

The manuscript titled “Developing an Advanced PM2.5 Exposure Model in Lima, Peru”, by Vu et al., presents a PM2.5 exposure model that incorporates satellite remote sensing, meteorological and land use data to estimate daily PM2.5 measurements at 1 km resolution. The authors claim that the model is the first of its kind in South America, providing greater spatial and temporal coverage than previous studies conducted in Peru. The output of this model will reduce the sparsity of PM2.5 measurements and could be applied to support future epidemiological studies.

The work presents several weakness that must be solved before publication. Firstly, the variable importance shown in figure 4 is very strange. Looking at figure 6, the spatial distribution of PM2.5 in the study area, it is obvious that “elevation” and “distance to shoreline” are very relevant parameters of the distribution. It is worrisome that the model doesn´t capture it. How do the authors explain that the distance from shoreline did not contribute (mentioned in lines 380-384), as it is fairly obvious from the figures. Better analysis must be provided to discard a model problem.

Response: Explanation of potentially why elevation and distance to shoreline are not important predictors in the model is added to the discussion. Briefly, variation in the distribution of elevation is significantly different between the two monitoring networks, with JHU sites having less variation than SENAMHI sites. About 25% of the total ground measurements come from JHU, where PM2.5 concentrations are also relatively homogenous due to the interpolation of weekly measurements to daily estimates, which may explain why elevation might not have been a more important predictor in the model. Additionally, distance to shoreline did not improve the model R2 and had little importance in the model. Additionally, annual mean predictions from a model that included distance to shoreline did not look different from predictions made with the final model that did not contain distance to shoreline.

Secondly, the validation of the AOD values with well-established data from the AERONET network (Section 2.3 Satellite data) must be expanded and the details of the comparison clearly explained. It is worrisome that the only reference is from a congress abstract without peer-review. (It can be located at https://ehp.niehs.nih.gov/doi/10.1289/isesisee.2018.O03.03.05 and it mentions: “This abstract was presented at the ISES-ISEE 2018 Joint Annual Meeting and has not been peer reviewed”).

Response: More information about the validation of AOD values with AERONET was added to section 2.3 to clarify that the Arica site was compared to MAIAC AOD observations in a 5x5km box surrounding that site. Furthermore, more updated references are added to this section.

Finally, the future research must be reconsider, The authors express their intention to convert PM10 data into PM2.5 (lines 404-405). PM10 measurement can follow very different trends than PM2.5. depending on the size distribution of the aerosols, and it can lead to wrong conclusions. Also, before applying the model-derived dataset to epidemiological studies (line 413), the model must prove its forecasting capacity on daily basis. Such study was not attempted in this work and it should be the step ahead.

Response: Attempts to prove the model forecasting capacity on a daily basis is added as a future step at the end of the discussion. A study by Silva et al. (2018) had already established the relationship between PM10 and PM2.5 from the same monitoring stations in Lima with results showing good correlation between them. We plan to further explore this relationship with the hope of extending our methodology to other regions in South America.

Adding to these, some parts of the manuscript needed clarification, included as notes in the attached file. Taking all this into account, my recommendation is that a major revision is necessary.

Responses to Comments from the PDF

Line 22-23, description of acronyms taken out for RF and CV per reviewer’s request

Line 46-48, added a reference to support statement

Line 49-56, reviewer suggested combining this paragraph with section 2.1 à paragraph was rewritten to not repeat information from section 2.1.

Line 66, replaced “health outcomes” with “diseases” per reviewer’s request

Line 75-96, revised to clarify which studies used MAIAC AOD vs not MAIAC AOD per reviewer’s request

Line 148, added a reference for MAIAC algorithm description per reviewer’s request

Line 150-158, added information clarifying the validation of MAIAC AOD with Arica AERONET per reviewer’s request

Line 186, Reviewer commented that the equations used to calculate RH here is old and imprecise. The reviewer suggested using IAPWS-95 formulation to calculate RH. Although this is a great suggestion, the parameters used to calculated RH (temperature and dew point) from ECMWF were of course resolution, and RH was calculated at 12.5km2 spatial resolution before being interpolated down to 1km2. Therefore, such calculations may not have a big impact on the results.

Line 187, clarified what “personal” weather stations meant per reviewer’s request

Line 190, added a statement about why ground measurements of temperature and relative humidity was compared with WRF-Chem temperature and ECMWF relative humidity per reviewer’s request

Line 211, added a few sentences regarding the advantages of the RF model per reviewer’s request

Line 224-231, reviewer commented that cross-validation usually requires separation of dataset into three groups: training, testing, and CV. We used a 10-fold cross-validation method with only two groups: training and testing, which is the more commonly practiced procedure in these types of studies. Also, our dataset comprised of 8470 total observations, and and splitting such a sparse dataset into three subsets may impact cross-validation performance.

Line 235-239, the reviewer commented that elevation is obviously not normally distributed and why would we study it. Here, we are only merely pointing out the distribution of the elevation variable. Although not mentioned here, the distribution of elevation differs by monitoring network, which may affect model performance and prediction capabilities as mentioned in the discussion.

 Line 254, reviewer suggests that the study would be better if only SENAMHI data is employed. JHU data was added to the model to enhance both the spatial and temporal coverage of PM2.5 in Lima since SENAMHI stations themselves do not have enough spatial and temporal coverage to accurately train the model.

Line 278, reviewer emphasized same comment as line 254.

Line 298-301, reviewer commented that the statement made is speculative. Speculative portion of the statement was deleted.

 Figure 3, reviewer suggested adding units to axis on the Bland-Altman Plot. Units have been added.

Line 316-317, reviewer commented on the importance of elevation on Figure 6. A portion pertaining to elevation distribution from both networks have been added to the discussion section to address this comment.

Line 328-329, reviewer commented that monthly trends can be calculated from weekly data and interpolation to daily can be skipped; however, monthly estimates from Figure 5 was calculated from daily predictions using the model, which was why JHU weekly measurements were interpolated down to daily estimates in the first place.

Line 351, reviewer commented that the same reference is mentioned in 8 lines and to rephrase text to mention it only once. Paragraph has been revised to better summarize that study.

Line 381-384, reviewer asked how authors explain that distance from shoreline did not contribute to model. Distance to shoreline was added to the model after the first RF model predictions showed that there was a gradient of PM2.5 rising up to the mountains in line with elevation. To investigate this issue, distance to shoreline was added to the RF model, and a 10-fold cross validation procedure was carried out. There was no difference in R2 between RF models with and without distance to shore. Furthermore, annual prediction maps of the RF model containing distance to shoreline was identical to prediction maps from the RF model without distance to shoreline, indicating that this variable did not contribute greatly to the model.

Line 404-405, reviewer commented that PM10 to PM2.5 conversion is a misleading path to follow since PM10 measurement can follow different trends than PM2.5. Results from a study by Silva et al. (2018) was added to demonstrate the relationship between PM2.5 and PM10 in Lima and the feasibility of such future work.

Line410-414, reviewer suggests that model forecasting capacity on a daily basis must be proven before use in epidemiologic studies. This suggestion has been implemented in the future research direction in the discussion.

Reviewer 3 Report

Adverse health effects in relation to sparsity of PM2.5 measurements in South America are known and investigated here for Lima, Peru due to Lima’s topography and aging vehicular fleet. An advanced machine-learning model to estimate daily PM2.5 concentrations at a 1 km2 spatial resolution in Lima is developed from 2010 to 2016. The combined aerosol optical depth (AOD), meteorological fields from the European Centre for Medium-Range Weather Forecasts (ECMWF), parameters from the Weather Research and Forecasting model coupled with Chemistry (WRF-Chem), and land use variables are applied to fit a random forest (RF) model against ground measurements from 16 monitoring stations for ground-based PM2.5 concentrations. 94.5% of observations fall within 2 standard deviations of the difference indicating good agreement between ground measurements and predicted estimates. Surface downwards solar radiation, temperature, relative humidity, and AOD were the most important predictors, while percent urbanization, albedo, and cloud fraction were the least important predictors. Mean annual maps of PM2.5 show consistent lower concentrations in the coast and higher concentrations in the mountains, resulting from prevailing coastal winds blown from the Pacific Ocean in the west. Our model allows for construction of long-term historical daily PM2.5 measurements at 1 km2 spatial resolution to support future epidemiological studies.

General comments

It is concluded that the model allows the construction of long-term historical daily PM2.5 measurements at 1 km2 spatial resolution to support future epidemiological studies. These conclusions are well discussed.

The paper addresses relevant scientific questions. The paper presents novel concepts, ideas and tools.

The scientific methods and assumptions are valid and clearly outlined so that substantial conclusions are reached.

The description of experiments and calculations are sufficiently complete and precise to allow their reproduction by fellow scientists.

The quality, information and caption of the figures are good.

The related work is well cited.

Title and abstract reflect the whole content of the paper. The abstract should be focused more on the results of this study.

The overall presentation is well structured and clear. The language is fluent but can be improved in some parts.

The abbreviations and units are generally correctly defined and used.

Specific Comments

It would be helpful to show in chapter 2.2. the measurement principles, the references for this or the guidelines for the measurements which are used.

In chapter 2.4 it is necessary to give information about the emission inventory which is used for WRF Chem.

Technical corrections

Where can one find the explanation of the abbreviations in Fig. 2, Table 1 and Fig. 5? This should be part of the figure caption.

Author Response

Comments and Suggestions for Authors

Adverse health effects in relation to sparsity of PM2.5 measurements in South America are known and investigated here for Lima, Peru due to Lima’s topography and aging vehicular fleet. An advanced machine-learning model to estimate daily PM2.5 concentrations at a 1 km2 spatial resolution in Lima is developed from 2010 to 2016. The combined aerosol optical depth (AOD), meteorological fields from the European Centre for Medium-Range Weather Forecasts (ECMWF), parameters from the Weather Research and Forecasting model coupled with Chemistry (WRF-Chem), and land use variables are applied to fit a random forest (RF) model against ground measurements from 16 monitoring stations for ground-based PM2.5 concentrations. 94.5% of observations fall within 2 standard deviations of the difference indicating good agreement between ground measurements and predicted estimates. Surface downwards solar radiation, temperature, relative humidity, and AOD were the most important predictors, while percent urbanization, albedo, and cloud fraction were the least important predictors. Mean annual maps of PM2.5 show consistent lower concentrations in the coast and higher concentrations in the mountains, resulting from prevailing coastal winds blown from the Pacific Ocean in the west. Our model allows for construction of long-term historical daily PM2.5 measurements at 1 km2 spatial resolution to support future epidemiological studies.

General comments

·         It is concluded that the model allows the construction of long-term historical daily PM2.5 measurements at 1 km2 spatial resolution to support future epidemiological studies. These conclusions are well discussed.

·         The paper addresses relevant scientific questions. The paper presents novel concepts, ideas and tools.

·         The scientific methods and assumptions are valid and clearly outlined so that substantial conclusions are reached.

·         The description of experiments and calculations are sufficiently complete and precise to allow their reproduction by fellow scientists.

·         The quality, information and caption of the figures are good.

·         The related work is well cited.

·         Title and abstract reflect the whole content of the paper. The abstract should be focused more on the results of this study.

·         The overall presentation is well structured and clear. The language is fluent but can be improved in some parts.

·         The abbreviations and units are generally correctly defined and used.

Specific Comments

It would be helpful to show in chapter 2.2. the measurement principles, the references for this or the guidelines for the measurements which are used.

Response: Information about the equipment and calibration has been added to section 2.2.

In chapter 2.4 it is necessary to give information about the emission inventory which is used for WRF Chem.

Response: The emissions inventory was described previously in a paper by Sánchez-Ccoyllo et al. and this reference is cited.

Technical corrections

Where can one find the explanation of the abbreviations in Fig. 2, Table 1 and Fig. 5? This should be part of the figure caption.

Response: The abbreviations are names of the 10 monitoring stations from SENAMHI derived from their location. This description has been added to the caption of figure 2 and 5, and table 1.

Round 2

Reviewer 2 Report

The authors have addressed all my comments and improved the manuscript accordingly, therefore, it can be published in current form